

# A SNP variation in the Sucrose synthase (*SoSUS*) gene associated with sugar-related traits in sugarcane

Supaporn Khanbo[1], Suthasinee Somyong[1], Phakamas Phetchawang[1], Warodom Wirojsirasak[2], Kittipat Ukoskit[3], Peeraya Klomsa-ard[2], Wirulda Pootakham[1] and Sithichoke Tangphatsornruang[1]

[1] National Center for Genetic Engineering and Biotechnology (BIOTEC), National Science and Technology Development Agency (NSTDA), Pathum Thani, Thailand
[2] Mitr Phol Innovation and Research Center, Phu Khiao, Chaiyaphum, Thailand
[3] Department of Biotechnology, Faculty of Science and Technology, Thammasat University, Pathumtani, Thailand

Corresponding author
Sithichoke Tangphatsornruang,
sithichoke.tan@nstda.or.th

## ABSTRACT

**Background:** Sugarcane (*Saccharum* spp.) is an economically significant crop for both the sugar and biofuel industries. Breeding sugarcane cultivars with high-performance agronomic traits is the most effective approach for meeting the rising demand for sugar and biofuels. Molecular markers associated with relevant agronomic traits could drastically reduce the time and resources required to develop new sugarcane varieties. Previous sugarcane candidate gene association analyses have found single nucleotide polymorphism (SNP) markers associated with sugar-related traits. This study aims to validate these associated SNP markers of six genes, including *Lesion simulating disease 1* (*LSD*), *Calreticulin* (*CALR*), *Sucrose synthase 1* (*SUS1*), *DEAD-box ATP-dependent RNA helicase* (*RH*), *KANADI1* (*KAN1*), and *Sodium/hydrogen exchanger 7* (*NHX7*), in a diverse population in 2-year and two-location evaluations.

**Methods:** After genotyping of seven targeted SNP markers was performed by PCR Allelic Competitive Extension (PACE) SNP genotyping, the association with sugar-related traits and important cane yield component traits was determined on a set of 159 sugarcane genotypes. The marker-trait relationships were validated and identified by both t-test analysis and an association analysis based on the general linear model.

**Results:** The mSoSUS1_SNPCh10.T/C and mSoKAN1_SNPCh7.T/C markers that were designed from the *SUS1* and *KAN1* genes, respectively, showed significant associations with different amounts of sugar-related traits and yield components. The mSoSUS1_SNPCh10.T/C marker was found to have more significant association with sugar-related traits, including pol, CCS, brix, fiber and sugar yield, with *p* values of $6.08 \times 10^{-6}$ to $4.35 \times 10^{-2}$, as well as some cane yield component traits with *p* values of $1.61 \times 10^{-4}$ to $3.35 \times 10^{-2}$. The significant association is consistent across four environments.

**Conclusion:** Sucrose synthase (*SUS*) is considered a crucial enzyme involved in sucrose metabolism. This marker is a high potential functional marker that may be used in sugarcane breeding programs to select superior sugarcane with good fiber and high sugar contents.

# INTRODUCTION

Sugarcane (*Saccharum* spp.) is one of the most important economic crops, with widespread cultivation in more than 100 countries around the world. The crop is a major source of sugar, accounting for 80% of the world's sugar production, and is increasingly utilized to produce sustainable fuels (*You et al., 2021*). To increase sugarcane production in a sustainable way, breeding new sugarcane varieties for higher yield is essential (*Yang et al., 2020*). However, sugarcane has one of the most complex genomes of all crop plants due to its high ploidy level, high heterozygosity, frequent aneuploidy, and large genome size (approximately 10 Gb) (*Banerjee et al., 2020*; *Grivet & Arruda, 2002*; *Mancini et al., 2017*; *Thirugnanasambandam, Hoang & Henry, 2018*). Modern sugarcane cultivars (2n = 100–140) were derived from interspecific crosses between *S. officinarum* (2n = 8x = 80; x = 10) and *S. spontaneum* (2n = 5x − 16x = 40 − 128; x = 8), followed by several backcrosses with *S. officinarum*, contributing about 80% and 10–20% to the hybrid genome, respectively (*D'Hont et al., 1998*; *D'Hont, 2005*; *Grivet & Arruda, 2002*; *Shearman et al., 2022*). This complicated genome organization makes trait inheritance difficult to analyze and makes the time required to develop new varieties as high as 12–15 years by conventional breeding (*Fickett et al., 2019*; *Gouy et al., 2015*). Reducing the duration of the breeding cycle without sacrificing the precision of the breeding value of selected clones in each generation is a crucial goal of sugarcane breeding (*Satpathy et al., 2022*).

Marker-assisted selection (MAS) can overcome the limitations of conventional breeding and increase its efficiency. The use of molecular markers associated with desirable traits has the potential to improve the accuracy of the selection process and considerably reduce the time required for breeding programs (*Collard & Mackill, 2008*; *Xu & Crouch, 2008*). Molecular markers are valuable tools for genetic studies and plant breeding. Several types of molecular markers based on DNA polymorphism have been developed for sugarcane, including restriction fragment length polymorphism (RFLP) (*Dai & Long, 2015*), random amplified polymorphic DNA (RAPD) (*Singh et al., 2017*), amplified fragment length polymorphism (AFLP) (*Debibakas et al., 2014*), simple sequence repeat polymorphism (SSR) (*Singh et al., 2020*), and, more recently, single-nucleotide polymorphism (SNP) (*Molina et al., 2022*; *You et al., 2019*). SNPs have become the markers of choice for mapping and MAS because of their codominance, abundance, and decreased cost of sequencing, as well as the availability of robust SNP genotyping techniques (*Rasheed et al., 2017*; *Thomson et al., 2017*). In recent years, a large number of genetic loci and SNPs that contribute to important agronomic traits in sugarcane have been identified through different approaches such as quantitative trait loci (QTL) mapping, gene mapping, and genome-wide association studies (GWASs) (*Fickett et al., 2019*; *Islam et al., 2018*; *Khanbo et al., 2021*; *Yang et al., 2018*, *2020*). However, these reported SNPs need to be validated for the linked traits to use them as molecular markers for MAS. To facilitate their use for MAS in sugarcane breeding, some of the trait-linked SNPs have been converted into

allele-specific polymerase chain reaction assays (*Gao et al., 2022*). These SNP markers are valuable tools for marker-assisted sugarcane breeding.

Following the study by *Khanbo et al. (2021)* that identified seven SNP markers associated with three sugar-related traits, including commercially extractable sucrose (CCS), sucrose content (pol), and fiber, with phenotypic variance explained ranging from 3.79 to 12.21%. These associated SNP markers were located in six genes, including *Lesion simulating disease 1* (*LSD*), *Calreticulin* (*CALR*), *Sucrose synthase 1* (*SUS1*), *DEAD-box ATP-dependent RNA helicase* (*RH*), *KANADI1* (*KAN1*), and *Sodium/hydrogen exchanger 7* (*NHX7*), which are involved in signaling and transcriptional regulation. Given their reported association with sugar-related traits, these SNP markers were targeted for conversion to the PACE assay.

Therefore, our study aimed to develop and validate SNP PACE markers for sugar-related traits. These assays were evaluated using seventeen segregating populations and a diverse collection of sugarcane genotypes. The conversion of these SNP markers into PACE genotyping assays could greatly improve the speed and efficiency of selection in sugarcane breeding programs.

## MATERIALS AND METHODS

### Plant materials and phenotypic data

A total of 170 progenies derived from 17 biparental crosses were used to test individual SNP assays for screening polymorphic markers (Table S1). After this initial testing, a panel comprising 159 diverse sugarcane accessions was genotyped with polymorphic SNP assays for marker validation. This population comprised a wide range of clones (*Wirojsirasak et al., 2023*), including two from Argentina, 13 from Australia, four from Barbados, three from Brazil, two from China, one from Cuba, seven from Fiji, one from Guyana, six from India, four from Mauritius, three from the Philippines, one from Reunion, three from South Africa, one from Sri Lanka, 10 from Taiwan, 85 from Thailand, and nine from the USA, as well as four of unknown origin (Table S2). Field experiments for the population were conducted at two locations: the Agronomy Field Crop Station, Faculty of Agriculture, Khon Kaen University, and the Mitr Phol Innovation and Research Center, Phu Khieo, Chaiyaphum, during the 2017–2018 (plant cane) and 2018–2019 (first ratoon) cropping seasons. For these locations, the diverse population was planted in randomized complete blocks with two replicates. Plot length was 5 m with 1.65 m between-row spacing.

Phenotypic data were collected following *Wirojsirasak et al. (2023)*. Briefly, the data were measured at 12 months after planting for the six sugar-related traits and six cane yield components, which included soluble solid content (Brix), sucrose content (Pol), commercially extractable sucrose (CCS), fiber percentage (Fiber), sugar yield, fiber yield, cane diameter, cane weight, cane length, node length, number of nodes, and cane yield. For all traits, measurements were performed on six stalks for each replicate. Shapiro-Wilks tests for normality were performed on all tested traits using SPSS Statistics version 16.0 (SPSS® Software GmbH, Munich, Germany). The same software was used to analyze the maximum, minimum, average, and standard deviation. The analysis of variance (ANOVA) was performed using SAS Studio (https://welcome.oda.sas.com/). To explore the

relationships among the studied traits across the cropping seasons, we used the average values of the two replicates for pairwise correlation calculations based on the Spearman method. Additionally, we generated the correlation matrix plot using the "GGally" package (https://CRAN.R-project.org/package=GGally) in R studio version 4.1.3 (*R Core Team, 2018*). The genetic variance and error variance from the ANOVA were used to estimate the broad sense heritability ($H^2$) for each trait. Broad sense heritability was estimated using the formula: $H^2 = \sigma_g^2/(\sigma_g^2 + \sigma_e^2/r)$ where $\sigma_g^2$ is the genetic variance, $\sigma_e^2$ is the residual variance, and r is the number of replicates (*Holland, Nyquist & Cervantes-Martínez, 2003*).

## DNA extraction

Genomic DNA was isolated from leaf tissues of the sugarcane clones using a DNA extraction kit, DNeasy Plant Mini Kit (QIAGEN, Germantown, MD, USA) and using the cetyltrimethyl ammonium bromide (CTAB) method (*Gawel & Jarret, 1991*). The quantity and quality of the total DNA were evaluated using 1% agarose gel electrophoresis and a NanoDrop™ 1000 Spectrophotometer (Thermo Fisher Scientific, Waltham, MA, USA), respectively.

## SNP Marker designing and SNP genotyping

To validate the SNP markers associated with sugar-related traits from a previous study (*Khanbo et al., 2021*; *Piriyapongsa et al., 2018*), a total of 7 SNP markers were converted to the PACE assay (Table S3), which comprised 2 SNPs for *LSD1* gene, one SNP for *CALR* gene, one SNP for *SUS1* gene, one SNP for *RH* gene, one SNP for *KAN1* gene and one SNP for *NHX7* gene. For marker assay development, the 250 bp upstream and downstream sequences of the selected SNPs were extracted from the genomic sequences of the sugarcane reference genome (*Garsmeur et al., 2018*). The sequences were used to design primers specific to the targeted SNPs (Table S3). SNP primers were designed by 3CR Bioscience Co. Ltd., Harlow, Essex, UK. The PACE SNP genotyping method followed the procedure previously described in *Somyong et al. (2022)*. The assay consists of two allele-specific forward primers with unique tail sequences that correspond with one of the two universal fluorescent resonance energy transfer (FRET) cassettes and the targeted SNP at the 3′ end, and one common reverse primer. Biallelic discrimination of the PACE assay was achieved through the competitive binding of two allele-specific forward primers. The Universal Genotyping Master Mix (Standard ROX) (Essex CM20 2BU; 3CR Bioscience, Harlow, UK) contained the solution required for PCR amplification and fluorescent detection, including FAM (blue emission) and HEX (red emission) specified for each SNP change. For genotyping, the reaction mix contained 2.5 µl Genotyping Master Mix, 0.07 µl assay mix, and 2.5 µl of 1–10 ng genomic DNA. The Genotyping Master Mix contained *Taq* DNA polymerase, universal fluorescent reporting cassette, dNTPs, performance enhancers, $MgCl_2$, and 5-carboxy-x-rhodamine, succinimidyl ester (ROX). The assay mix contained allele-specific primer1-FAM, allele-specific allele2-HEX and common reverse primer with a final concentration ratio of 0.168: 0.168: 0.42 µM, respectively, for each reaction. PCR was run initially for 15 min at 94 °C, followed by 10 cycles for 20 s at 94 °C and 1 min at 61 °C to 55 °C in touchdown mode with a decrease of 0.6 °C in every cycle,

and finally 25 cycles of 20 s at 94 °C and for 1 min at 57 °C, with an ending step of 60 s at 37 °C. The QuantStudio 6 Flex real-time PCR system (Thermo Fisher Scientific, Waltham, MA, USA) was utilized for carrying out the real-time PCR reaction. The data was then analyzed using QuantStudio™ Real-Time PCR Software v1.3 to identify SNP genotypes. After evaluation of the polymorphisms in the progeny population, only polymorphic PACE markers were used for genotyping the diverse population that had phenotypic data.

## Statistical and association analyses

To determine preliminary relationships of polymorphic loci with sugar-related traits and cane yield components, the data were analyzed using independent sample t-tests in SPSS Statistics version 16.0 (SPSS® Software GmbH, Munich, Germany) and boxplots were created using the "ggplot2" package (https://CRAN.R-project.org/package=ggplot2) in R studio version 4.1.3. (*R Core Team, 2018*). To validate the significant relationship between the targeted SNP markers and the traits, we performed an association analysis using TASSEL version 5.2.8 (*Bradbury et al., 2007*). This analysis was based on a simple general linear model (GLM) without covariate control and included 10,000 permutations. The required data for the association analysis included genotype and phenotype information. Non-normally distributed phenotype data were transformed (logarithmic, square or square root) to normality prior to analysis. Significance of marker-trait association was determined with a threshold $p$-value of less than 0.05.

# RESULTS

## Phenotypic evaluation

A diverse collection of 159 sugarcane accessions was evaluated in four environments (2017–2018KKU, 2018–2019KKU, 2017–2018MPT, and 2018–2019MPT) for a total of 12 traits, comprising six sugar-related traits and six cane yield components. The Shapiro-Wilk test revealed that some traits were not normally distributed at $p < 0.01$ (Table S4). All traits exhibited a wide range of phenotypic variation among the accessions (Table S5).
For example, in the 2017–2018KKU environment, brix ranged from 14.95 to 23.85%, with an average of 19.65%; pol ranged from 7.76 to 21.84%, with an average of 15.85%; CCS ranged from 4.36 to 16.66%, with an average of 11.25%; and cane diameter ranged from 1.27 to 3.5 cm with an average of 2.57 cm. The frequency distribution histograms of all traits across the four environments displayed a continuous pattern, suggesting the quantitative nature of the inheritance of these traits (Figs. S1 and S2). Analysis of variance revealed significant differences ($p$ value $< 0.01$) among genotypes for all traits. The broad sense heritability ($H^2$) estimates of all traits in each environment were low to high, ranging from 19.38 (cane yield in 2018–2019KKU) to 97.35% (brix in 2018–2019MPT). The high level of $H^2$ indicates less environmental influence on the expression of characteristics (Table S5).

A Spearman's correlation analysis was conducted to explore pairwise relationships among the studied traits. The correlations among the six sugar-related traits across four environments that derived from two years (2017–2018 and 2018–2019) and two locations (KKU and MPT) are depicted in Fig. S3. Within the 2017–2018KKU environment, a total

of 12 significant correlations were identified ($p < 0.05$), consisting of seven positive and five negative correlations. The pairs of CCS-pol, brix-pol, brix-CCS, and sugar yield-fiber yield showed highly positive correlations, ranging from 0.69 to 0.97 ($p < 0.001$), while fiber was significantly negatively correlated with CCS ($p < 0.01$). These trends were consistent in the other three environments. Furthermore, when analyzing the correlation between the same traits across the years 2018 and 2019, significantly high correlations were observed in both the KKU and MPT locations (Fig. S3).

For the cane yield components, in the 2017–2018KKU environment, we identified 13 significant correlation trait-pairs ($p < 0.05$) among all traits. Of these pairs, 11 were positively correlated, and two exhibited negative correlations. Most of the relationships between traits showed significant correlations. For instance, the cane weight traits showed positive correlations with cane diameter, cane length, number of nodes and cane yield. In addition, cane length showed significant positive correlations with node length, number of nodes, and cane yield, while node length was negatively correlated with number of nodes and cane diameter. The highest correlation was between cane weight and cane diameter traits ($p < 0.001$). Likewise, in the other three environments, the analysis of pairwise correlations between traits exhibited a pattern similar to that observed in the 2017–2018KKU environment. The correlation between the same traits across the years 2018 and 2019 showed significantly high correlations in both locations (Fig. S4).

## High through-put genotyping by PACE SNP genotyping

The primary objective of this research was to develop SNP markers, which were genotyped using SNP genotyping. A total of seven primer sets targeting six genes (*LSD1*, *CALR*, *SUS1*, *RH*, *KAN1*, and *NHX7*) were designed and listed in Table S6. These primer sets, including mSoLSD1_SNPch7.G/T, mSoLSD1_SNPch7.C/A, mSoCALR_SNPch2.A/G, mSoSUS1_SNPCh10.T/C, mSoRH56_SNPCh3.T/C, mSoKAN1_SNPCh7.T/C, and mSoNHX7_SNPCh8.A/C, were developed. Among these primer sets, only two (mSoSUS1_SNPCh10.T/C and mSoKAN1_SNPCh7.T/C) were found to be polymorphic in the progeny population, while the rest (mSoLSD1_SNPch7.G/T, mSoLSD1_SNPch7.C/A, mSoCALR_SNPch2.A/G, mSoRH56_SNPCh3.T/C, and mSoNHX7_SNPCh8.A/C) were monomorphic (Fig. S5). For marker validation, the diverse sugarcane population was genotyped with two polymorphic SNP assays. The genotypes of the polymorphic markers were presented in Fig. 1 and Table S7, where mSoSUS1_SNPCh10.T/C had two genotypes (T/T and T/C) in 120 and 35 individuals, respectively (Figs. 1A and 1B), and mSoKAN1_SNPCh7.T/C had two genotypes (T/T and T/C) in 116 and 40 individuals, respectively (Figs. 1C and 1D). Furthermore, the distribution of allele frequencies in the homozygous and heterozygous groups was observed for the two polymorphic markers.

## Statistical analysis

The t-test and association analysis of the traits with SNP markers were tested at two locations (KKU and MPT) during the 2017–2018 and 2018–2019 cropping seasons. Data from each environment (2017–2018KKU, 2018–2019KKU, 2017–2018MPT, and 2018–2019MPT) were used to confirm the significant association of SNP markers with

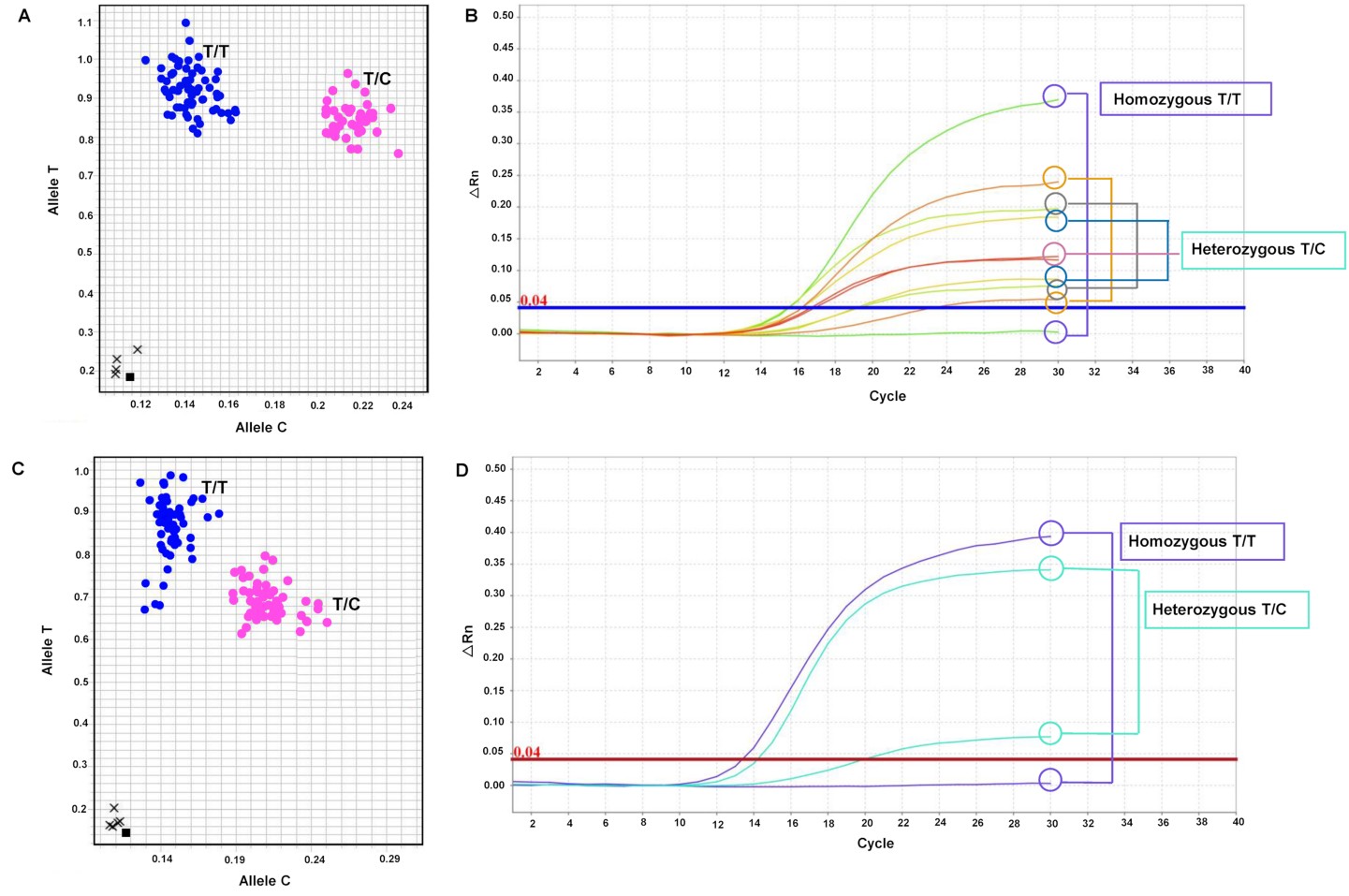

**Figure 1 Allelic discrimination plots and amplification plots of the polymorphic SNP markers.** (A and B) mSoSUS1_SNPCh10.T/C (blue: genotypes T/T and pink: T/C), and (C and D) mSoKAN1_SNPCh7.T/C (blue: genotypes T/T and pink: T/C).

each trait. The t-test analysis that was conducted on sugar-related traits revealed significant differences between the T/T and T/C genotypes of the mSoSUS1_SNPCh10.T/C marker for pol, CCS, and fiber in all four environments, with $p$ values ranging from 0.000 to 0.045 (Table S8 and Fig. 2). Furthermore, the mSoSUS1_SNPCh10.T/C genotypes were found to have a significant brix difference in the 2017–2018KKU environment, and these genotypes exhibited significant sugar yield differences in both the 2017–2018MPT and 2018–2019MPT environments. Sugarcane individuals with the T/T genotype demonstrated significantly higher levels of pol, CCS, brix and sugar yield when these individuals were compared with the T/C genotype individuals. Conversely, sugarcane individuals with the T/C genotype exhibited significantly higher levels of fiber compared with those individuals with the T/T genotype (Fig. 2).

In addition to the sugar-related traits, the mSoSUS1_SNPCh10.T/C marker was also found to show significant differences between the T/T and T/C genotypes for cane yield component traits. Specifically, significant differences were observed in cane diameter

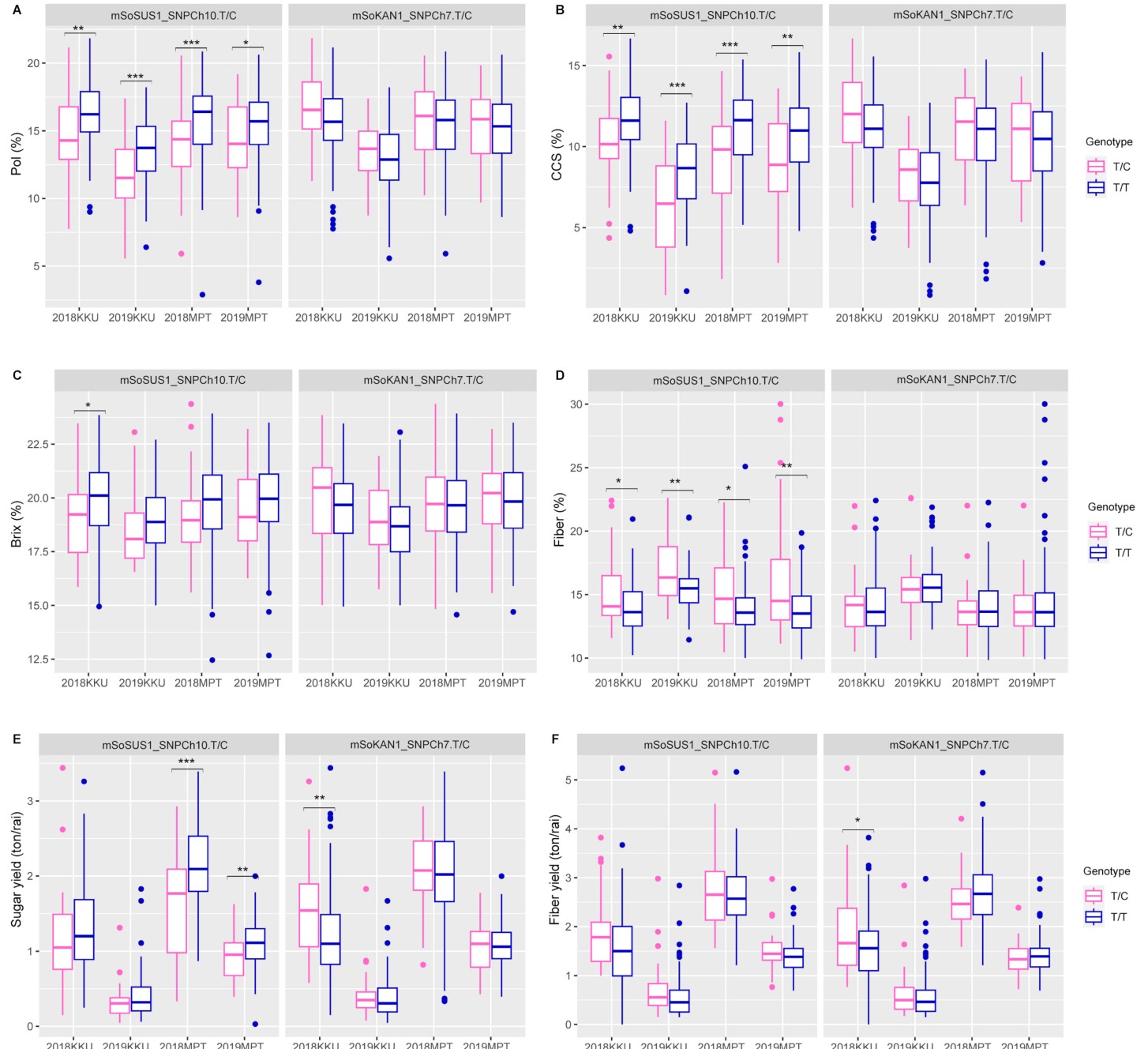

**Figure 2 Boxplots showing the effect of the SNP markers, including mSoSUS1_SNPCh10.T/C and mSoKAN1_SNPCh7.T/C associated with sugar-related traits.** (A) Pol, (B) CCS, (C) Brix, (D) fiber, (E) sugar yield, (F) fiber yield, 2018, 2017–2018 cropping season (plant cane); 2019, 2018–2019 cropping season (first ratoon); KKU, Khon Kaen; MPT, Mitr Phol Innovation and Research Center. $^*p < 0.05$; $^{**}p < 0.01$; $^{***}p < 0.001$

across all four environments, as shown in Table S8 and Fig. 3. Additionally, significant differences were observed in cane weight across three environments, with the exception of the 2018-2019KKU. Furthermore, the T/T and T/C genotypes exhibited significant differences in cane length and node length in the 2018–2019KKU and 2017–2018KKU
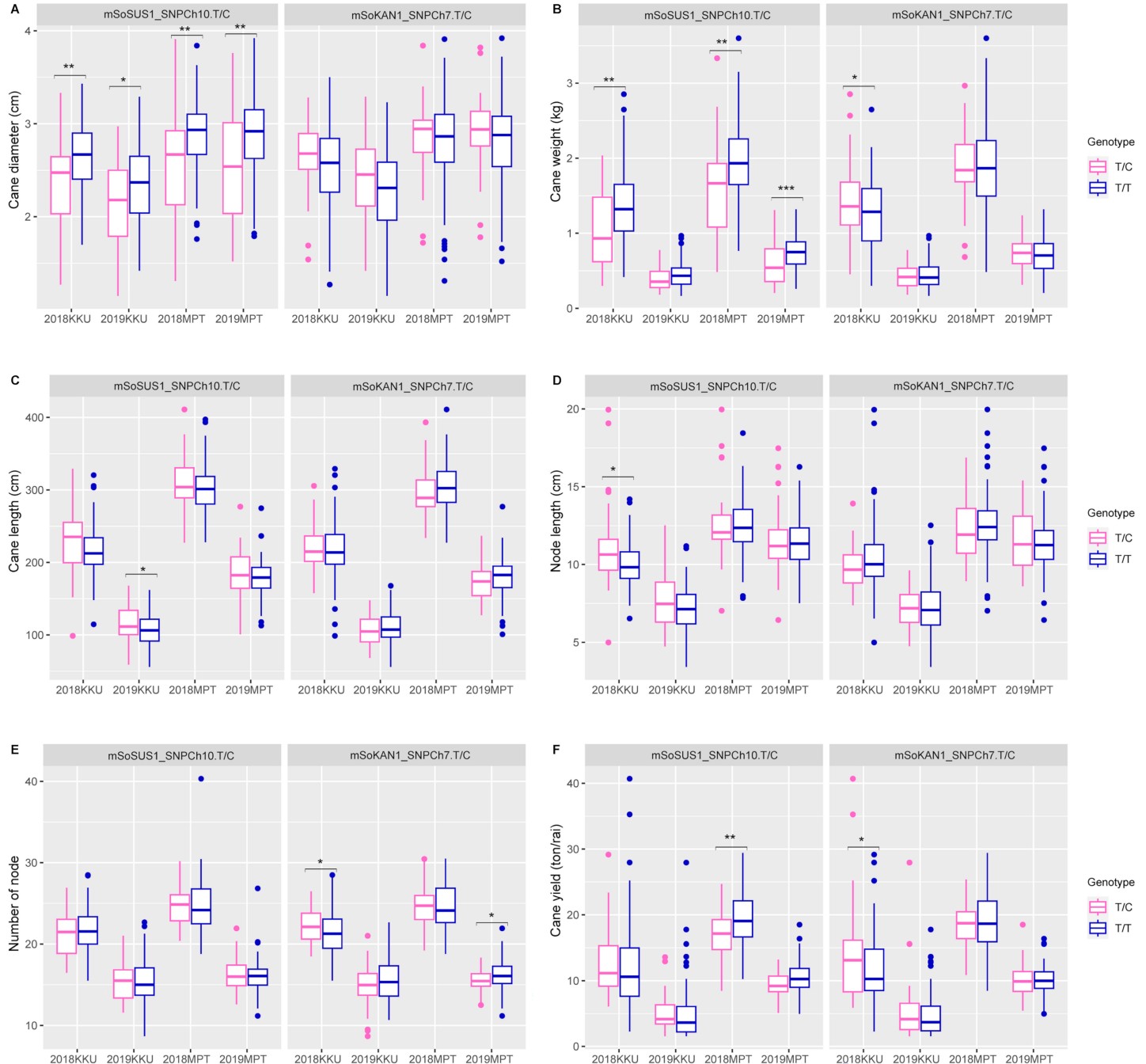

**Figure 3 Boxplots showing the effect of the SNP markers, including mSoSUS1_SNPCh10.T/C and mSoKAN1_SNPCh7.T/C associated with cane yield components.** (A) Cane diameter, (B) cane weight, (C) cane length, (D) node length, (E) number of nodes, and (F) cane yield. 2018, 2017–2018 cropping season (plant cane); 2019, 2018–2019 cropping season (first ratoon); KKU, Khon Kaen; MPT, Mitr Phol Innovation and Research Center. *$p < 0.05$; **$p < 0.01$; ***$p < 0.001$.

environments, respectively, as well as significant differences in cane yield in the 2017–2018MPT environment. Sugarcane individuals with the T/T genotype showed significantly higher levels of cane diameter, cane weight, and cane yield in comparison to those with the T/C genotype. Conversely, the individuals with the T/C genotype displayed

significantly higher levels of cane length and node length compared with those individuals with the T/T genotype (Fig. 3).

For the mSoKAN1_SNPCh7.T/C marker, significant differences were observed between the T/T and T/C genotypes for sugar-related traits. These differences were identified for sugar yield and fiber yield in the 2017–2018KKU environment, with $p$ values ranging from 0.003 to 0.035 (Table S8 and Fig. 2). Additionally, the mSoKAN1_SNPCh7.T/C genotypes showed significant differences in cane weight and cane yield in the 2017–018KKU environment, and significant differences in the number of nodes in both the 2017–2018 KKU and 2018-2019MPT environments (Table S8 and Fig. 3). Sugarcane individuals with the T/C genotype exhibited significantly higher levels of sugar yield, fiber yield, cane weight, number of nodes, and cane yield than those individuals with the T/T genotype. Conversely, in 2018–2019MPT environment, the individuals with the T/T genotype had significantly higher levels of the number of nodes compared to those individuals with the T/C genotype (Fig. 3).

The GLM model was employed in the study for association mapping analysis. The results revealed that the mSoSUS1_SNPCh10.T/C marker was significantly associated ($p < 0.05$) with pol, CCS, and fiber, of sugarcane grown in all four environments as shown in Table 1. Moreover, this marker was significantly linked ($p < 0.05$) with the cane diameter of sugarcane grown in all four environments and linked with cane weight in three environments. Additionally, the marker was also found to be significantly associated ($p < 0.05$) with cane length of sugarcane grown only in the KKU environments.

Another marker, mSoKAN1_SNPCh7.T/C, displayed a statistically significant correlation ($p < 0.05$) with sugar yield, cane weight, and cane yield of sugarcane grown in the 2017–2018 KKUenvironment. Moreover, in the 2018–2019MPT environment, the marker was found to be significantly associated ($p < 0.05$) with the number of nodes. However, the mSoKAN1_SNPCh7.T/C marker did not significantly associate with any traits of sugarcane grown in the 2018–2019KKU and 2017–2018MPT environments. These findings are summarized in Table 1. These association results support data from the t-test analysis.

## DISCUSSION

Identification of loci influencing sugar related-traits and yield components facilitates marker-assisted breeding to increase crop productivity. Several other sugarcane studies have identified marker-trait associations (MTAs) for sugar-related traits and cane yield components (*Banerjee et al., 2015*; *Fickett et al., 2019*; *Khanbo et al., 2021*; *Siraree et al., 2017*; *Ukoskit et al., 2019*). Validation of those markers in populations with different genetic backgrounds and environmental conditions is essential for their successful application in breeding programs. In this work, PACE markers were developed, based on previously identified markers from candidate gene association mapping for sugar-related traits (*Khanbo et al., 2021*). The associations between these markers and traits were validated using phenotype data that exhibited a satisfactory level of variation in phenotypic traits, comparable to data from other studies (*Banerjee et al., 2015*; *Fickett et al., 2019*; *Gouy et al., 2015*; *Khanbo et al., 2021*). The broad phenotypic range observed for each trait

**Table 1 Markers associated with sugar-related traits and cane yield components of the diverse population with $p < 0.05$ in 2017–2018 (plant cane) and 2018–2019 (first ratoon) cropping seasons at the KKU and MPT locations.**

| Crop cycle | Location | Trait | mSoSUS1_SNPCh10.T/C | | | | mSoKAN1_SNPCh7.T/C | | | |
|---|---|---|---|---|---|---|---|---|---|---|
| | | | Allele | $p$ value | $R^2$ | Functional annotation | Allele | $p$ value | $R^2$ | Functional annotation |
| 2017–2018 (Plant cane) | KKU | Pol | T/C | **4.30E−04** | 7.91 | Sucrose synthase 1 | T/C | 0.06475 | 2.21 | Transcription repressor KAN1 |
| | | CCS | | **0.00255** | 5.99 | | | 0.11611 | 1.64 | |
| | | Brix | | **0.04166** | 2.72 | | | 0.18693 | 1.14 | |
| | | Fiber | | **0.00748** | 4.80 | | | 0.91005 | 0.01 | |
| | | Sugar yield | | 0.20236 | 1.09 | | | **0.00237** | 5.96 | |
| | | Fiber yield | | 0.07085 | 2.22 | | | 0.06749 | 2.26 | |
| | | Cane diameter | | **2.85E−04** | 8.38 | | | 0.20293 | 1.06 | |
| | | Cane weight | | **0.00217** | 6.06 | | | **0.03509** | 2.87 | |
| | | Cane length | | **0.02885** | 3.12 | | | 0.88343 | 0.01 | |
| | | Node length | | 0.06286 | 2.30 | | | 0.21868 | 1.00 | |
| | | Number of nodes | | 0.3087 | 0.69 | | | 0.06853 | 2.15 | |
| | | Cane yield | | 0.41282 | 0.45 | | | **0.03633** | 2.87 | |
| 2018–2019 (1st Ratoon) | KKU | Pol | T/C | **5.41E−04** | 10.09 | Sucrose synthase 1 | T/C | 0.32139 | 0.86 | Transcription repressor KAN1 |
| | | CCS | | **5.39E−05** | 13.49 | | | 0.27143 | 1.05 | |
| | | Brix | | 0.33436 | 0.83 | | | 0.67125 | 0.16 | |
| | | Fiber | | **1.96E−04** | 12.21 | | | 0.74278 | 0.10 | |
| | | Sugar yield | | 0.19247 | 1.28 | | | 0.49401 | 0.35 | |
| | | Fiber yield | | 0.07216 | 2.49 | | | 0.8645 | 0.02 | |
| | | Cane diameter | | **0.03354** | 3.38 | | | 0.2046 | 1.20 | |
| | | Cane weight | | 0.17655 | 1.39 | | | 0.44272 | 0.44 | |
| | | Cane length | | **0.02523** | 3.74 | | | 0.17804 | 1.35 | |
| | | Node length | | 0.06312 | 2.59 | | | 0.77619 | 0.06 | |
| | | Number of nodes | | 0.81557 | 0.04 | | | 0.1746 | 1.37 | |
| | | Cane yield | | 0.25692 | 0.97 | | | 0.40246 | 0.52 | |
| 2017–2018 (Plant cane) | MPT | Pol | T/C | **0.00124** | 6.61 | Sucrose synthase 1 | T/C | 0.39737 | 0.47 | Transcription repressor KAN1 |
| | | CCS | | **1.25E−04** | 9.26 | | | 0.25028 | 0.86 | |
| | | Brix | | 0.08905 | 1.88 | | | 0.97462 | 0.00 | |
| | | Fiber | | **0.00978** | 4.40 | | | 0.45857 | 0.37 | |
| | | Sugar yield | | **6.08E−06** | 12.71 | | | 0.6018 | 0.18 | |
| | | Fiber yield | | 0.27901 | 0.77 | | | 0.18635 | 1.13 | |
| | | Cane diameter | | **1.61E−04** | 8.92 | | | 0.5062 | 0.29 | |
| | | Cane weight | | **2.84E−04** | 8.28 | | | 0.64724 | 0.14 | |
| | | Cane length | | 0.3357 | 0.61 | | | 0.22606 | 0.95 | |
| | | Node length | | 0.75159 | 0.07 | | | 0.44968 | 0.37 | |
| | | Number of nodes | | 0.67652 | 0.11 | | | 0.98981 | 0.00 | |
| | | Cane yield | | 0.00803 | 4.50 | | | 0.74023 | 0.07 | |

*(Continued)*

| Table 1 (continued) | | | | | | | | | | | |
|---|---|---|---|---|---|---|---|---|---|---|---|
| Crop cycle | Location | Trait | mSoSUS1_SNPCh10.T/C | | | | mSoKAN1_SNPCh7.T/C | | | | |
| | | | Allele | p value | R² | Functional annotation | Allele | p value | R² | Functional annotation | |
| **2018–2019** (1ˢᵗ Ratoon) | MPT | Pol | T/C | **0.04354** | 2.64 | Sucrose synthase 1 | T/C | 0.73679 | 0.07 | Transcription repressor KAN1 | |
| | | CCS | | **0.00234** | 5.93 | | | 0.64697 | 0.14 | | |
| | | Brix | | 0.26986 | 0.80 | | | 0.89559 | 0.01 | | |
| | | Fiber | | **1.03E–04** | 9.41 | | | 0.36489 | 0.53 | | |
| | | Sugar yield | | **0.0023** | 5.95 | | | 0.99157 | 0.00 | | |
| | | Fiber yield | | 0.08825 | 1.90 | | | 0.30173 | 0.70 | | |
| | | Cane diameter | | **0.00209** | 6.02 | | | 0.28243 | 0.75 | | |
| | | Cane weight | | **5.05E–04** | 7.68 | | | 0.47818 | 0.33 | | |
| | | Cane length | | 0.28806 | 0.74 | | | 0.12968 | 1.49 | | |
| | | Node length | | 0.55838 | 0.22 | | | 0.74208 | 0.07 | | |
| | | Number of nodes | | 0.61287 | 0.17 | | | **0.01986** | 3.47 | | |
| | | Cane yield | | 0.08078 | 1.99 | | | 0.85718 | 0.02 | | |

**Note:**
Bold values indicate statistical significance at the $p < 0.05$ level. $R^2$ = Percentage of phenotypic variance explained by the marker. KKU, Khon Kaen; MPT, Mitr phol innovation and research centre.

reflects the considerable genetic variability within the population (Table S5). To determine the relationship between genetic and environmental influence on trait variation, the broad-sense heritability of each trait was calculated (Table S5). The moderate-to-high heritability estimated for most of the phenotypic traits indicates that a substantial proportion of the observed variation is attributable to genetic inheritance. These heritability estimates fall within the range reported in other sugarcane studies (*Barreto et al., 2019*; *Fickett et al., 2019*; *Gouy et al., 2015*; *Yang et al., 2020*). The heritability of a phenotypic trait plays a crucial role in determining the effectiveness of SNP identification in association mapping.

The association analysis revealed multiple significant association between SNP markers and the sugar-related traits, as well as cane yield components (Table 1). Some of these associations were consistent across environments, while some were specific to certain environments. This could be because these traits are influenced by multiple genes and exhibit significant genotype-by-environment interactions. Interestingly, mSoSUS1_SNPCh10.T/C consistently associated with pol, CCS, and fiber across all environments (Table 1). These results suggest that mSoSUS1_SNPCh10.T/C exhibit genetic stability, making it a reliable and promising candidate for future utilization in marker-assisted selection (MAS) programs within sugarcane breeding.

Most of the traits in sugarcane are quantitative, multigenic and/or multiallelic in nature (*Banerjee et al., 2015*; *Meena et al., 2022*). Analysis of phenotypic data for twelve quantitative traits in sugarcane varieties revealed significant positive correlations among traits such as brix, pol, CCS, and sugar yield, while negative correlations were observed between CCS and fiber across different environments (Fig. S3). A high degree of

correlation between sucrose traits was reported in previous studies (*Fickett et al., 2019*; *Khanbo et al., 2021*; *Siraree et al., 2017*). Additionally, consistent with earlier research (*Banerjee et al., 2015*; *Barreto et al., 2019*), a strong positive correlation was observed between cane yield components, including cane diameter and cane weight, as well as cane length and node length (Fig. S4). Thus, the identification of marker-trait associations (MTAs) for any of these traits may also hold potential as MTAs for other correlated traits. The mSoSUS1_SNPCh10.T/C and mSoKAN1_SNPCh7.T/C markers showed multiple significant associations with sugar-related traits and some cane yield components. These results may be attributed to the loci being located in or linked to pleiotropic genes, or to the biochemical interrelationships among these traits (*Yang et al., 2020*). Utilizing these pleiotropic markers would facilitate the simultaneous selection of multiple sugar-related traits.

According to the analysis of the effects of different genotypes on sugar-related traits in the population, the mSoSUS1_SNPCh10.T/C marker was significantly associated with CCS and fiber, exhibiting inverse genotype effects between these two traits (Figs. 2B and 2D). This pattern is consistent with the high negative correlation observed between CCS and fiber (Fig. S3). Within the sugarcane population, the mSoSUS1_SNPCh10.T/C marker exhibits two genotypes: TT and TC. The polymorphic alleles (C and T) in mSoSUS1_SNPCh10.T/C contributes to a change in sugar content. The T allele is associated with high sugar contents, while the C allele has the opposite effect, associating with decreased sugar levels (Figs. 2A–2C). Conversely, the C allele in the marker is associated with an increase in fiber content (Fig. 2D). The findings of the current study show that mSoSUS1_SNPCh10.T/C might play an important role in affecting sugar-related traits as well as cane yield components and could be linked to genes that affect these traits in sugarcane.

Many SNPs significantly associated with the traits of interest were nonsynonymous, meaning they altered the amino acid sequence and directly influenced gene function (*Parida et al., 2016*). However, in the present study, the two significant SNPs identified were located in introns, which are noncoding regions of the gene. These findings are consistent with the results of *Lu et al. (2018)*, who also found SNPs associated with trait variance in *CfSUS* located in introns or synonymous mutations. Additionally, *Porth et al. (2013)* reported that most SNPs significantly associated with traits were found in noncoding regions. Although synonymous mutations or mutations in introns and noncoding regions do not cause changes in amino acids, they can still contribute to trait variation through other mechanisms (*Stern & Orgogozo, 2008*). The association between SNPs in noncoding regions and phenotypic variation can be explained by two hypotheses. Firstly, the identified SNPs may be in linkage disequilibrium (LD) with causal polymorphisms located in nearby genes or regulatory sequences responsible for the observed trait variance (*Stam & Laurie, 1996*). Secondly, nucleotide alterations within introns could play a role in post-transcriptional regulation, such as alternative splicing (*Szakonyi & Duque, 2018*; *Wang et al., 2017*).

The mSoSUS1_SNPCh10.T/C marker is located within the *sucrose synthase 1 (SUS1)* gene. In our study, we identified association between allelic variation in *SUS1* and

sugar-related traits, including CCS and fiber. These findings support the significant role of *SUS1* in influencing sugar and fiber content, consistent with previous studies (*Khanbo et al., 2021*). Sucrose synthase is a key enzyme involved in sucrose metabolism and is closely related to sucrose content (*Khan et al., 2021*). It catalyzes the reversible conversion of sucrose into fructose and UDP-glucose, which are primary substrates for respiration, starch synthesis, and cell wall constituents (*Gerber et al., 2014*; *Koch, 2004*). Moreover, sucrose synthase likely contributes to fiber development by providing hexose components for cellulose synthesis (*Zeng et al., 2016*). SUS activity is prominent in the young internodes of sugarcane stems, positively correlated with hexose sugars, and negatively correlated with sucrose concentration (*Goldner, Thom & Maretzki, 1991*; *Verma et al., 2011*). Furthermore, *SUS* expression levels are higher in high sugar genotypes compared to low sugar genotypes, with increasing SUS activity correlating with enhanced sucrose accumulation rate and ripening (*Lingle & Irvine, 1994*). However, other studies have reported no correlation between sucrose synthase activity and sucrose accumulation (*Botha & Black, 2000*; *Zhu, Komor & Moore, 1997*). The specific involvement of SUS in sucrose accumulation in sugarcane is not fully understood (*Anur et al., 2020*).

In cucumber, downregulation of the *CsSUS4* gene resulted in reduced cellulose, starch, and hexose contents, negatively impacting flower and fruit growth (*Fan et al., 2018*). Additionally, overexpression of the potato *SUS* gene in cotton increased fiber production (*Xu et al., 2012*). These findings suggest that *SUS* plays a significant role in determining sucrose content and fiber development.

Another marker, the mSoKAN1_SNPCh7.T/C, is located within the *KANADI1* (*KAN1*) gene, which belongs to the GARP family of plant transcription factors, plays a crucial role in governing abaxial identity, leaf growth, and meristem formation in *Arabidopsis thaliana* (*Huang et al., 2014*). KAN1 has also been reported to repress the transcription of AS2 (ASYMMETRIC LEAVES2) in abaxial cells and regulate adaxial-abaxial polarity (*Wu et al., 2008*). It participates in the regulation of various physiological processes in plants, including embryo, shoot, and root patterning, while also governing multiple organ polarity processes during vegetative growth (*Emery et al., 2003*; *Eshed et al., 2004*; *Hawker & Bowman, 2004*; *Ilegems et al., 2010*; *Izhaki & Bowman, 2007*; *Merelo et al., 2013*). In this study, a significant association was observed between the mSoKAN1_SNPCh7.T/C marker and some traits under certain environmental conditions, suggesting that this gene may indirectly influence sugar production in sugarcane by regulating the expression of genes involved in the development of sugar-rich tissues.

## CONCLUSIONS

In conclusion, PACE markers were developed based on previously identified markers from association mapping and subsequently validated in a diverse population. Among these markers, the mSoSUS1_SNPCh10.T/C marker exhibited a significant association with sugar-related traits and cane yield components. The consistency of the results across diverse genetic backgrounds and different environmental conditions indicates the substantial potential and reliability of the marker for future marker-assisted selection (MAS) in sugarcane breeding programs.

## ACKNOWLEDGEMENTS

We would like to thank Mr. Derrick Thompson for reading and thoroughly checking English usage and grammar.

### Funding

This research has received funding support from the NSRF *via* the Program Management Unit for Human Resources & Institutional Development, Research and Innovation (PMU-B) [grant number B01F640054] and National Science and Technology Development Agency (grant number P-2050223). The funders had no role in study design, data collection and analysis, decision to publish, or preparation of the manuscript.

### Grant Disclosures

The following grant information was disclosed by the authors:
Human Resources & Institutional Development, Research and Innovation (PMU-B): B01F640054.
National Science and Technology Development Agency: P-2050223.

### Competing Interests

The authors declare that they have no competing interests. Warodom Wirojsirasak, and Peeraya Klomsa-ard are employed by Mitr Phol Sugarcane Research Center Co., Ltd.

### Author Contributions

- Supaporn Khanbo conceived and designed the experiments, performed the experiments, analyzed the data, prepared figures and/or tables, authored or reviewed drafts of the article, and approved the final draft.
- Suthasinee Somyong conceived and designed the experiments, analyzed the data, prepared figures and/or tables, authored or reviewed drafts of the article, and approved the final draft.
- Phakamas Phetchawang performed the experiments, prepared figures and/or tables, and approved the final draft.
- Warodom Wirojsirasak performed the experiments, analyzed the data, prepared figures and/or tables, and approved the final draft.
- Kittipat Ukoskit conceived and designed the experiments, authored or reviewed drafts of the article, and approved the final draft.
- Peeraya Klomsa-ard performed the experiments, prepared figures and/or tables, and approved the final draft.
- Wirulda Pootakham conceived and designed the experiments, authored or reviewed drafts of the article, and approved the final draft.
- Sithichoke Tangphatsornruang conceived and designed the experiments, analyzed the data, authored or reviewed drafts of the article, and approved the final draft.

## Data Availability

The genotype data are available in the Supplemental File.

## Supplemental Information

Supplemental information for this article can be found online at http://dx.doi.org/10.7717/peerj.16667#supplemental-information.

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
