# Peer review of "A SNP variation in the Sucrose synthase (SoSUS) gene associated with sugar-related traits in sugarcane"

_PeerJ, doi:10.7717/peerj.16667_

## Round 0.1 · original submission · Major Revisions

Dear dr Tangphatsornruang,

Regarding your submitted manuscript: A SNP variation in the Sucrose synthase (SoSUS) gene affects sugar-related traits in sugarcane.

Two reviewers have completed their assessment and believe that your manuscript can be accepted after revision. Besides these comments, I would appreciate it if you could take into consideration some remarks of my own.

It is important in order for research to be replicated and confirmed, to present M&M more analytically.
• Hence, R version, suites used, and the code should be presented or cited.
• Moreover, the pipeline used for tassel is not very clear.
• Finally, in terms of data analysis it would be interesting to present correlations across the studied traits among the two years (for instance as a figure). This would provide information on whether association is biased or not on a specific year, and pinpoint outliers.

Looking forward for the revised version,

Dr Nikolaos Nikoloudakis

**Language Note:** The review process has identified that the English language must be improved. PeerJ can provide language editing services - please contact us at copyediting@peerj.com for pricing (be sure to provide your manuscript number and title). Alternatively, you should make your own arrangements to improve the language quality and provide details in your response letter. – PeerJ Staff

Reviewer 1 ·

Basic reporting

Khanbo et al. validated a SNP variation in the Sucrose synthase (SoSUS) gene associated sugar-related traits in sugarcane Their results suggest that this SNP marker could serve as a dependable tool in sugarcane breeding programs.

My major concerns list as follows:

1. In line 87, it would be better to spell out the full name of these genes.

2. Please supply titles for all of your supplementary Tables.

3. In line 181, it should be changed to "(Fig. S1-S4)" to avoid confusion.

4. In lines 188-195, the text is a bit challenging to follow. I recommend adding the source of your conclusion at the end of each sentence, indicating which figure or table supports the statement. This will make it clearer for readers to understand your findings.

5. In line 176, it seems unusual that Table S4 was not mentioned, but Table S5 was referenced first. Please ensure that the tables are mentioned in the correct order to avoid confusion for the readers.

6. For Figure S1-S5, it would be helpful to include figure legends that provide clear explanations and details about each figure. The current versions of the figures are confusing.

7. In lines 250-252, your statement is not accurate. In 2017-2018 KKU. It seems that the individuals with the T/C genotype had significantly higher levels of the number of nodes compared to those individuals with the T/T genotype (Fig. 3).

8. When discussing your findings in the discussion section, it's essential to include the source of your statements at the end of each sentence, specifying which figure or table provides support for the statement. For instance, the statements in line 303, lines 308-311, line 324, and lines 328-329 can be challenging for readers to track without clear references.

Experimental design

9. In lines 179-181, in fact, for some traits like SY and CY, they may not follow a normal distribution.

10. In line 187, you should conduct a Shapiro-Wilk test for normality and then decide which correlation method (Pearson or Spearman) to use based on the results.

Validity of the findings

11. The title is inappropriate as the study only found an association between a SNP variation in the Sucrose synthase (SoSUS) gene and sugar-related traits without confirming a direct causal effect on these traits.

Reviewer 2 ·

Basic reporting

The English language needs to improve more. Many statements have missed references. Abbreviation needs to be defined at first appearance. The discussion section needs to be rewritten since many of the statements are duplicates from the M&M and results section.

Experimental design

Adequate and perfect

Validity of the findings

The findings were supported by the results.

---

## Round 0.2 · accepted · Accept

The authors have done a thorough revision and addressed all concerns. in my opinion, the manuscript can be accepted as it stands.

Reviewer 1 ·

Basic reporting

The authors have made thorough revisions to the manuscript. However, please ensure that the figure legends for all supplementary figures are provided before the publication.

Experimental design

NA

Validity of the findings

NA